# Seasonal Proteome Variations in *Orbicella faveolata* Reveal Molecular Thermal Stress Adaptations

**DOI:** 10.3390/proteomes12030020

**Published:** 2024-07-10

**Authors:** Martha Ricaurte, Nikolaos V. Schizas, Ernesto F. Weil, Pawel Ciborowski, Nawal M. Boukli

**Affiliations:** 1Department of Marine Sciences, University of Puerto Rico, Mayagüez, Call Box 9000, Mayagüez, PR 00681, USA; martha.ricaurte@upr.edu (M.R.);; 2Mass Spectrometry and Proteomics Core Facility, Durham Research Center, University of Nebraska Medical Center, Omaha, NE 68198, USA; 3Biomedical Proteomics Facility, Microbiology and Immunology Department, Universidad Central del Caribe, Bayamón, PR 00960, USA

**Keywords:** seasonal fluctuations, *Orbicella faveolata*, proteomics, thermal stress response

## Abstract

Although seasonal water temperatures typically fluctuate by less than 4 °C across most tropical reefs, sustained heat stress with an increase of even 1 °C can alter and destabilize metabolic and physiological coral functions, leading to losses of coral reefs worldwide. The Caribbean region provides a natural experimental design to study how corals respond physiologically throughout the year. While characterized by warm temperatures and precipitation, there is a significant seasonal component with relative cooler and drier conditions during the months of January to February and warmer and wetter conditions during September and October. We conducted a comparative abundance of differentially expressed proteins with two contrasting temperatures during the cold and warm seasons of 2014 and 2015 in *Orbicella faveolata*, one of the most important and affected reef-building corals of the Caribbean. All presented proteoforms (42) were found to be significant in our proteomics differential expression analysis and classified based on their gene ontology. The results were accomplished by a combination of two-dimensional gel electrophoresis (2DE) to separate and visualize proteins and mass spectrometry (MS) for protein identification. To validate the differentially expressed proteins of *Orbicella faveolata* at the transcription level, qRT-PCR was performed. Our data indicated that a 3.1 °C increase in temperature in *O. faveolata* between the cold and warm seasons in San Cristobal and Enrique reefs of southwestern Puerto Rico was enough to affect the expression of a significant number of proteins associated with oxidative and heat stress responses, metabolism, immunity, and apoptosis. This research extends our knowledge into the mechanistic response of *O. faveolata* to mitigate thermal seasonal temperature variations in coral reefs.

## 1. Introduction

The Caribbean is home to 9% of the world’s coral reefs, but only about one-sixth of the original area of coral cover remains [1]. Coral reefs face multiple stressors that threaten to damage coral reefs throughout the Caribbean region, including anthropogenic factors, ocean acidification, global warming, bleaching, diseases, and rising sea temperatures. Diseases have caused profound changes in the Caribbean coral reefs in the past 30 years, resulting in very few disease-free areas, including reefs. *Orbicella faveolata*, commonly known as volcanic star coral, is one of the founding coral species of the Caribbean that has exhibited a significant decline in recent decades, including in many shallow reef areas highly exposed to UV light [2,3,4,5,6,7]. These stressors have led to substantial deterioration of coral reefs and have subsequently resulted in corals’ adaption to heat stress, both physiologically and biochemically [8,9,10]. The duration of exposure to high temperatures above the thermal tolerance of corals is one of the main adaptations needed for corals to resist and survive temperature changes.

Puerto Rico is one of the regions in the Caribbean that experiences two seasons, namely, cold and warm seasons. The temperature fluctuations between 26.7 °C in the cold season and 29.8 °C in the warm are ideal conditions to study how corals cope with these natural changes. Recently, many of the studies on the effects of environmental stresses on corals conducted around the world have focused on how corals respond to thermal stress. For example, a previous finding highlighted that the reef-building coral *Acropora palmata* mitigates its thermal stress by upregulating proteins, such as heat shock proteins and green fluorescent proteins, as a mechanism against oxidative attack by reactive oxygen species (ROS) [11]. Previous studies have indicated that ambient and elevated water temperatures cause hypoxia in *O. faveolata*, reducing oxygen consumption [12]. Additionally, researchers have used artificial intelligence to predict bleaching in *O. faveolata* based on protein signatures at higher temperatures [13]. Further studies examining the proteomes of both coral hosts (*Orbicella faveolata*) and their endosymbiotic dinoflagellates under different laboratory temperatures found increased protein turnover at higher temperatures [14]. However, understanding *O. faveolata* coral differential protein expression response and how the species adapts to warmer seasonal temperatures conditions (from cold to warm seasons) has not been studied. This work represents the first comparative proteomics analysis of proteoforms derived from *O. faveolata* during the cold and warm seasons in two different reefs—the inner-shelf reef Enrique and the mid-shelf reef San Cristobal, southwest Puerto Rico, in the Caribbean.

Taken together, our results show that *O. faveolata*, like other scleractinian corals, possesses the capacity to mitigate thermal seasonal variations. This capacity may directly impact the metabolic and stress responses of *O. faveolata*, enhancing its ability to cope with fluctuating temperatures.

## 2. Materials and Methods

### 2.1. Study Area and Sampling Locations

This study was conducted at the inner-shelf reef Enrique (17°57.15′ N, 67°02.55′ W) and the mid-shelf reef San Cristobal (17°56.30′ N, 67°04.45′ W) in La Parguera Puerto Rico (Figure 1). Twelve colonies of *O. faveolata* were collected by SCUBA divers at depths ranging from 1 to 3 m, ensuring a 3 m distance between sampling locations to minimize sampling genetic clones [15,16,17]. The colonies were tagged by inserting a nail into the substrate near the colony and attaching a plastic tag with a unique identification number to the stainless steel nail. This allowed for the identification and location of the specific colonies again during 2014 and 2015 (see Appendix A). Monthly average sea surface temperatures (SSTs) were measured every hour in situ with a Hobo-pro during the cold (Jan-Feb) and warm (September–October) seasons of 2014–2015.

The collected colonies (3 × 3 cm) were placed in seawater and transported to the laboratory in a cooler bag. Three replicas of each fragment (2 cm^2^) were ground in 1 mL of rehydration buffer (9.5 M urea, 2% CHAPS, 1% DTT) in a mortar, following the published methodology [11]. The homogenized tissue was then lysed under ice sonication and stirred slowly at room temperature for 1 h. The samples were centrifuged at 12,000 rpm for 30 min at 4 °C, and the supernatant (without zooxanthellae localized in the middle layer after centrifugation) was transferred to a new 1.5 mL Eppendorf tube. Protein concentration was estimated using the Bradford method [18], following the manufacturer’s recommendations (Invitrogen, Carlsbad, CA, USA).

### 2.2. High-Resolution Two-Dimensional Gel Electrophoresis (2D-GE)

*O. faveolata*’s proteins (200 µg) were prepared for isoelectric focusing (IEF) in rehydration/sample buffer (Bio-Rad, Hercules, CA, USA) on 11 cm pH 3–10 immobilized pH gradient (IPG) strips. Upon termination of the IEF in a first dimension Bio-Rad protein cell (20,000 Vh/50 μA/strip) [11], the IPG strips were incubated in equilibration buffer for 2 × 15 min with reducing DTT agents and iodoacetamide alkylation, as described by Ricaurte and Boukli (2016). *O. faveolata* coral protein lysates (100 µg) were separated by gradient SDS polyacrylamide gels (4–20%). Protein spots were visualized using G-250 BioSafe Coomassie blue in gel staining, as described previously [11], and revealed from 2D-GE samples of the inner-shelf reef Enrique and the mid-shelf reef San Cristobal.

### 2.3. Gel Image and Statistical Analysis

Proteoforms were visualized through in-gel staining via 2D-GE and analyzed using a Bio-Rad imaging system (Versa Doc Model 1000). Experiments were performed in triplicate (144 2D-GE), and the resulting gels were analyzed in a single matchset using Bio-Rad PDQuest™ software (version 8.0.1). Following automatic spot detection, the files underwent manual inspection to evaluate the accuracy of the computer-generated images. Individual spot “volumes” were computed in each gel through density/area integration. Density variability among spots from different gels was controlled by normalization utilizing the total volume data of all the spots of *Orbicella faveolata*. Forty-two 2D-GE normalized spot volume instances were identified as significantly differing in abundance by ≥2-fold (Appendix A).

To improve the accuracy and quantification of protein levels across various gels, we standardized the spot volume density obtained from each coral lysate by normalizing it against the total spot volume of the respective gel using PDQuest software [11].

### 2.4. Protein in-Gel Digestion and Data Analysis

Coomassie blue-stained 2D-GE spots were manually excised. These gel fragments were then washed with Milli-Q water and treated with 40% methanol/10% acetic acid for destaining. They were subsequently alkylated with 55 mmol L^−1^ iodoacetamide in 50 mmol L^−1^ AB and reduced with 10 mmol L^−1^ DTT in 50 mmol L^−1^ ammonium bicarbonate (AB). The destaining solution was removed, and gel spots underwent three washes with 50 mmol L^−1^ AB and 50% acetonitrile (ACN). The samples were then digested with 5% proteomics grade (*w*/*w*) trypsin (Sigma-Aldrich, St. Louis, MO, USA) overnight at 37 °C. The digested samples were then mixed with an equal volume of saturated cyano-4-hydroxycinnamic acid in 50% ACN/0.1% trifluoracetic acid. Half of the resulting mixture from each spot was then applied to a matrix-assisted laser desorption/ionization (MALDI target plate).

### 2.5. Matrix-Assisted Laser Desorption/Ionization Time-of-Flight Mass Spectrometry (MALDI-TOF/TOF MS)

Mass spectrometric analysis was performed using a MALDI-TOF/TOF 4800 mass spectrometer obtained from Applied Biosystems (Foster City, CA, USA). A volume of one microliter of the digested spot from *O. faveolata* was applied directly onto the MALDI plate (Applied Biosystems) and combined with α-cyano-4-hydroxycinnamic acid (CHCA) matrix (5 mg/mL in 50% ACN-0.1% trifluoroacetic acid [TFA]) and air-dried. The 4000 series Explorer software (Applied Biosystems) version 3.5.1 allowed processing of the generated spectra by positive reflection mode. A mixture of five external standards (Applied Biosystems) was allowed to perform external calibration. Masses of the resulting peptide were generated in increments of 50 spectra, covering a mass range between 800 and 3000 Da. MS/MS spectra were acquired in 1 kV positive mode, and 1000 shots accumulated in increments of 50. The MatrixScience website (http://www.matrixscience.com/xmlns/schema/mascot_search_results, accessed on 15 December 2014), allowed database searches for the identified proteoforms from *O. faveolata*. Moreover, Protein Pilot search engine parameters were applied against the Swiss-Prot databank (www.expasy.org, accessed on 15 January 2014). The search parameters included carbamidomethylation for cysteines and oxidation for methionines as variable modifications, allowing for one missed tryptic cleavage, with a mass accuracy tolerance set at 100 ppm for precursors and 0.5 Da for fragments.

### 2.6. RNA Extraction

Live tissue samples were collected from three representative colonies of *O. faveolata* to extract total RNA by directly applying 1 mL of TRIzol^®^ (Invitrogen) to each fragment (~2 cm^2^) of coral surface as a modification of the manufacturer’s protocol [19]. All *O. faveolata* fragments from both cold and warm seasons were submerged in TRIzol^®^ solution in beakers and subjected to washing on a gyratory shaker for 1 h. After washing in TRIzol^®^, the process served as the homogenization step as per the manufacturer’s instructions. The TRIzol^®^ washed tissues were then portioned into 1 mL aliquots, and the RNA extraction procedure was carried out for each replicate/sample. RNA concentrations were estimated by excitable fluorescence at 485 nm (RiboGreen, Molecular Probes, Portland, OR, USA). The integrity of the total RNA was confirmed by electrophoresis of a sample aliquot on a 1% formaldehyde agarose gel [20]. Subsequently, the total RNA was further purified by DNase I digestion and extraction by phenol/CHCl_3_.

### 2.7. Primer Design and Reverse Transcription-Quantitative Polymerase Chain Reaction (RT-qPCR) Validation

Five qRT-PCR primers were designed using Primer 3 software (v.0.4.0) and dual-labeled with a fluorescent probe, such as SYBR Green. The reverse transcription target genes were chosen based on their expression and cellular function in *O. faveolata*, such as alpha carbonic anhydrase, cytochrome c oxidase subunit I, green fluorescent protein, beta-tubulin, and cysteine-rich protein (Table 1). The RNA (2 μg) from *O. faveolata* was reverse-transcribed using SsoAdvancedTM Universal SYBR^®^ Green Supermix (Bio-Rad) with the corresponding primers (Table 1). Semi-quantitative RT-qPCR was performed on 20 μL^−1^ reaction volumes using a PCR Master Mix (Bio-Rad) in a Master Cycler (Eppendorf, Hamburg, Germany) and the following cycling parameters: 98 °C for 30 s, 35 cycles of 98 °C for 15 s, 60 °C for 30 s, and 60 °C for 30 s. The reaction products were separated on 2% agarose gels. After confirming that each primer pair amplified only a single PCR product of the expected size, qRT-PCR was performed using a CFX96 TouchTM Real Time System (Bio-Rad). Reactions were performed in triplicate using a 20 μL^−1^ reaction volume with 2 μL^−1^ of cDNA SYBR Green PCR Master Mix (Qiagen, Hilden, Germany) and 30 pmoL^−1^ of each primer. qPCR conditions were 50 °C/2 min and 95 °C/10 min, followed by 40 cycles of 95 °C/15 s for denaturation and 60 °C/30 s for annealing and extension.

The qPCR data from all five genes (alpha carbonic anhydrase, cytochrome c oxidase subunit I, green fluorescent protein, beta-tubulin, and cysteine-rich protein) were transformed and normalized using the CV analysis and pairwise ΔCt method [25]. For accurate and robust normalization, three expressed reference genes (actin, ribokinase, and calcium-dependent protein-kinase) were required [26]. The actin gene was the most stable gene during the cold and warm seasons. This delta-delta-Ct (ΔΔCt) model involves a calculation of ΔCt with stability values (M) and coefficient of variations (CV) lower than 1 and 50%, respectively. All the data analysis was developed using a *t*-test with InfoStat statistical software (version 2014) (Appendix A).

## 3. Results

### 3.1. Upregulated and Downregulated Proteoforms

The *O. faveolata* colonies collected in 2014 from the inner-shelf reef Enrique revealed 580 differentially expressed proteoforms in both seasons. The 331 upregulated differentially expressed proteins in the inner-shelf reef Enrique were overexpressed compared with the cold season (CWCS), while the 249 downregulated genes had a lower expression compared with the warm season (CWWS), and 139 proteins were commonly expressed in that same reef. In the mid-shelf reef San Cristobal, the 318 upregulated differentially expressed proteins were overexpressed compared with the cold season (CWCS), while the 306 downregulated proteins had a lower expression compared with the warm season (CWWS), and 106 common proteins in that same reef. Similar results were obtained in 2015 (Figure 2, Table 2, and Appendix A).

### 3.2. Orbicella faveolata Proteoforms Altered under Cold and Warm Seasonal Stress Conditions

Forty-two proteoforms were selected by PDQuest™ software based on ≥2.0-fold changes (Figure 3). This subset was classified into 11 categories based on the gene ontology (GO) classification according to their main biological function. Overall, the highest percentage of upregulated proteoforms (CWCS) in the inner-shelf Enrique corresponded to heat stress proteins (i.e., heat shock protein), immune system process (i.e., toll-like receptor 2), and signal transducer activity and transcriptional regulation, among others (i.e., alpha i G protein and zinc finger protein 768-like (*Orbicella faveolata*), by 18%. On the other hand, the downregulated proteoforms (CWWS) appeared to be involved in the metabolism function (i.e., cytochrome oxidase subunit I, cytochrome P450 74 A) and the calcification process, among other processes (i.e., small cysteine rich protein 6) by 21%. This was followed by sensory perception (i.e., integrin beta, actin), apoptosis (i.e., caspase 8), and calcification (i.e., carbonic anhydrase 5 A and calmodulin) by 14%. The last 7% were UV-related (i.e., GFP-like fluorescent chromoprotein cFP484) and transcription factors (i.e., Pax C7) (Table 2, Figure 4A,B).

With respect to the mid-shelf reef San Cristobal, the highest percentage of upregulated proteoforms (CWCS) had a specific role in transcription and polypeptide binding (i.e., alpha q G protein and zinc finger protein 768-like (*Orbicella faveolata*) by 20%, followed by heat stress (i.e., heat shock transcription factor) and immunity system process (i.e., apextrin) by 17%. The three functions of oxidative stress (i.e., activin 1), sensory perception (i.e., actin, beta-tubulin), and UV-related factors (i.e., GFP-like fluorescent chromoprotein CFP484) were expressed by 10%. The downregulated proteoforms (CWWS) appeared to be involved in metabolism (i.e., cytochrome oxidase subunit I) by 25%, followed by sensory perception (i.e., gelsolin and myosin), apoptosis (i.e., caspase 8), miscellaneous (i.e., procollagen), and calcification (i.e., calmodulin) by 17% (Table 2, and Figure 4C,D).

### 3.3. Statistical Analysis

All experiments were performed with 12 biological samples during the cold season of 2014 in both reefs. These colonies were tagged to be sampled again during the warm season of 2014 and the next two seasons in 2015 for a total of 144 technical replicates (Appendix A). Image analysis and differentially expressed protein quantification results were analyzed using student *t*-tests performed at a 95% significance level to examine whether the differences in the detected proteoforms in *O. faveolata* were a function of seasonal temperature conditions (cold and warm seasons) during the years 2014 and 2015. Significant levels of expression (i.e., ≥2.0-fold) were tested using a Shapiro–Wilk test at a significance level of *p* < 0.05. The criteria for normality were met in all proteins expressed by *O. faveolata*. Analysis of variance (ANOVA), standardized skewness, and the Shapiro–Wilk test revealed that the data were statistically normal and there was normality between the variances with a significance level of *p* < 0.05. However, there were no significant differences between the years 2014 and 2015 (with ±95% confidence interval) (Appendix A).

To verify the reliability and reproducibility of our MS results, RT-qPCR was used to confirm the presence and relative abundance of mRNA transcripts corresponding to the proteins. The qPCR data from all five genes were transformed and normalized according to the methods proposed by Hellemans and colleagues, with M (stability values) and CV (coefficient of variations) lower than 1 and 50%, respectively [25]. The ΔΔCt model was used to analyze the relative quantification of the real-time PCR (Appendix A). Differences in gene expression were assessed with linear regression using the two temperature conditions of the cold and warm seasons.

## 4. Discussion

The proteoform profile of *O. faveolata* with seasonal temperature changes was enriched by eleven biological processes that displayed differential expression patterns (Figure 3) during the cold and warm seasons. One of the most significant altered functions was related to heat stress in both reefs (Figure 4). These proteoforms include heat shock proteins that interact with the respiratory process to maintain an energy source to be subsequently used for calcification [27,28]. An increase in respiratory rates under light that is associated with raised temperature could influence coral growth by giving signals to increase the amount of electron transport for energy production [29,30]. Additionally, the upregulation of the apextrin, toll-like receptors 2, and ATP synthase subunit 6, known to compromise immune defense for heat stress [31,32,33], suggests a potential role of these proteins in the coral *O. faveolata*’s protection mechanism. Taken as a whole, the range of proteomics data validated by RT-PCR analyses indicates that temperature tolerance is likely to give *O. faveolata* the ability to mitigate thermal seasonal temperature variations, especially during warmer months.

One of the most significant advantages for *O. faveolata* was its capacity to maintain its vital functions with the fluctuation of temperature between seasons. This ability was reflected in the upregulation of heat shock, activin 1, thioredoxin, beta integrin, and cysteine-rich proteoforms during elevated temperatures, protecting essential cellular components of *O. faveolata* from various types of damage harmful or even more severe stress. This suggests that *O. faveolata* incorporates oxygen into the extracellular matrix during thermal stress to prevent suffocation and acclimatizes physiologically [34,35,36]. Moreover, cytoskeletal proteoforms known to be involved in sensory perception and phagocytosis were downregulated, which may indicate that these mechanisms mediate the reduction of energy consumption in *O. faveolata* to regulate its metabolism when temperature rises. Additionally, downregulation of pro-apoptotic markers such as caspase 8 and ectonucleoside triphosphate diphosphohydrolase promoted inactivation of the signaling complex of cell survival. Consequently, this triggers activation of proteoforms, such as procollagen, carbonic anhydrase, and beta integrin, which are known to regulate cell signaling pathways in a calcium-dependent manner [37]. Indeed, experimental and field studies suggest that calcification has been shown to play a role in coral colony integrity for survival [38,39,40].

There is compelling evidence that actin is a protein that functions with myosin in the synthesis and calcification of the coral organic matrix [41]. These proteins also interact with cadherin proteins [42], reinforcing their role as potential markers and central regulators of protein-mediated intracellular transport, plasma membrane interactions, and cell shape and integrity in *O. faveolata* [38,39,40]. Additionally, green fluorescent proteins and inositol deacylase were upregulated in *O. faveolata* (Table 2). These proteins are capable of pronounced changes in their spectral properties in response to irradiation and in their capacity to adapt to UV radiation. This favors their ability to repair heat damage by subsequently helping to reduce coral stress [43,44] (Figure 4C,D). Mayfield and colleagues [45] reported similar results for stress-protective proteins, including those associated with thermotolerance, protein folding, calcification functions, and response to oxidative stress in *O. faveolata*, in a thermal laboratory-based setting mimicking conditions from inshore and offshore localities in south Florida. Interestingly, our results across experiments showed that temperature conditions (from the cold and warm seasons) in both reefs—the inner-shelf reef Enrique and the mid-shelf reef San Cristobal, southwest Puerto Rico—also led to key thermotolerance signature proteins that can be a valuable tool for future comparative studies on the identification of proteins and mechanisms that *O faveolata* employs to mitigate thermal stress.

### 4.1. Responsiveness to Stress in Orbicella faveolata

Exposure to elevated temperature intensifies coral stress, resulting in changes in the metabolic state of the coral. Failure of the *O. faveolata* normal cellular processes at a transcriptional/posttranslational level can cause the upregulation of key proteins released by the mitochondria or the ribosome structures. One of these differentially expressed proteins is cytochrome P450, known to be associated with mitochondrial function [46,47]. Other upregulated molecular proteoforms identified, which are well known to mitigate oxidative stress, such as ATP synthase, apextrin, and toll-like receptors, can also be involved in the first line of a marine organism’s physiological defense [47] (Table 2).

On the other hand, ribosomal proteins start consuming a large proportion of cellular energy [48,49] by inducing oxidative stress through mitochondrial cytochrome oxidase subunit 1 (COI) upregulation and by downregulating beta tubulins, which control the entry to the mitochondria [22,50]. These two proteins play a key role in coping with stressful environmental changes in corals by interacting with cytoskeletal proteins and actively participating in mitochondrial regulation [49,51,52]. Additionally, the expression of green fluorescent proteoform could reflect a decrease in photoprotection as an indicator of coral health decline before any bleaching signs could be observed [53] (Figure 5).

### 4.2. Expression Profile of the Reference Genes in Orbicella faveolata

In this study, the expression profiles of the five reference genes used, namely alpha carbonic anhydrase, cytochrome c oxidase subunit I, green fluorescent protein, beta-tubulin, and cysteine-rich protein genes, showed consistency between mRNA levels and their corresponding post-translational/post-transcriptional expression levels in *O. faveolata* samples during the cold and warm seasons in both reefs: the inner-shelf reef Enrique and the mid-shelf reef San Cristobal (Figure 5). A real-time dissociation curve confirmed the presence of a specific PCR product in all amplification reactions. The relative quantities of the five genes were normalized using the quantities of actin as a housekeeping gene (HKG) based on a recognized reference gene from other organisms (Table 1).

Based on the results, actin was a suitable gene choice for normalizing target gene expression, as determined by real-time qRT-PCR, with a threshold cycle (CT) value of 21.68. The screening qRT-PCR results revealed high expression for alpha carbonic anhydrase (CT value 6.6) and beta tubulin (CT value 10.9), while cysteine (CT value 5.8), cytochrome (CT value 6.3), and green fluorescent protein (CT value 6.4) were highly downregulated during the hot wet season in the inner-shelf reef Enrique.

Green fluorescent protein (CT value 6.48), alpha carbonic anhydrase (CT value 13.72), and beta tubulin (CT value 23.36) were among the top highly upregulated genes in the mid-shelf reef San Cristobal, while the cysteine gene (CT value 3.34) and cytochrome (CT value 21.06) were the most highly downregulated (Figure 5).

## 5. Conclusions

There is a consequent need to characterize the thermal stress responses of corals to understand the molecular processes underpinning these responses. Here, a comparative proteomic approach was undertaken with the endangered Caribbean reef-builder *Orbicella faveolata* to unravel differential protein expression under abrupt thermal environmental change, such as an increase in temperature during seasonal temperature conditions (cold and warm seasons) during the years 2014 and 2015 in southwest Puerto Rico.

The proteins identified and quantified by mass spectrometry were found to be enriched in 11 biological processes, which were subsequently validated through qPCR. The mRNA levels confirmed the expression abundance of the proteoform profile in *O. faveolata*. Seasonal fluctuation changes in water temperatures generally remained within a range of less than 4 °C across most tropical reefs [54,55]. Taken together, our results imply that a 3.1 °C increase in temperature in *O. faveolata* between the cold and warm seasons in the Enrique and San Cristobal reefs was enough to lead to heat stress-responsive differentially expressed proteoforms. This confirms previous findings emphasizing that corals are sensitive to temperature changes as low as 1–2 °C [56,57,58,59,60].

This proteomic study has contributed to our understanding of the dysregulated pathway (Figure 6) underlying 42 proteoform alterations associated with metabolic and stress responses to cope with temperature variations. In addition, the differentially expressed proteins involved in metabolism, oxidative, and stress responses can be seen as signature markers of thermal tolerance for future comparative studies. Overall, this project strengthens our knowledge concerning climate change mechanisms and provides a reference for screening candidate marker proteins related to seasonal temperature sensibility of coral reefs.

## 6. Limitations

For this 2D LC-MS/MS-based study, one of the limitations is that numerous proteins found in *O. faveolata* have unknown functions. For example, the coral *Montipora capitata* has ~12% of the predicted gene models with no significant ortholog [61]. Moreover, certain coral genes lack an annotation, with limited proteins containing a known domain and labeled as dark proteins [62]. These proteins can regulate coral stress response; subsequently, targeting them is a key step in understanding their function that may be specific to coral thermal tolerance.

Another notable limitation of upstream proteomic sample preparation is the complexity of protein concentrations and the dynamic range of coral biological samples. Downstream analytical aspects, such as 2D-GE image visualization via staining detection methods and software-based image analysis, have made enormous gains in reproducibility over the course of the past two decades. In addition, most of the staining detection methods are mass-spectrometry-compatible, so that sequence coverages in the range of 25–50% are very common. This results in better separation and reproducibility in the 2D-GE.

Thus, the main remaining factor for variability to study the proteome of *O. faveolata* under temperature seasonality fluctuations lies in the sample itself. Moreover, the iso-electric focusing (IEF) dimension can be very sensitive to contaminants that interfere with the resolution of the electrophoretic separation of coral samples. Subsequently, protein extraction and solubilization must be adapted to each sample type. For the current study, we observed limited sample complexity, making the 2D-GE and MS a successful tool for studying *O. faveolata* at a proteomic scale.

## Figures and Tables

**Figure 1 proteomes-12-00020-f001:**
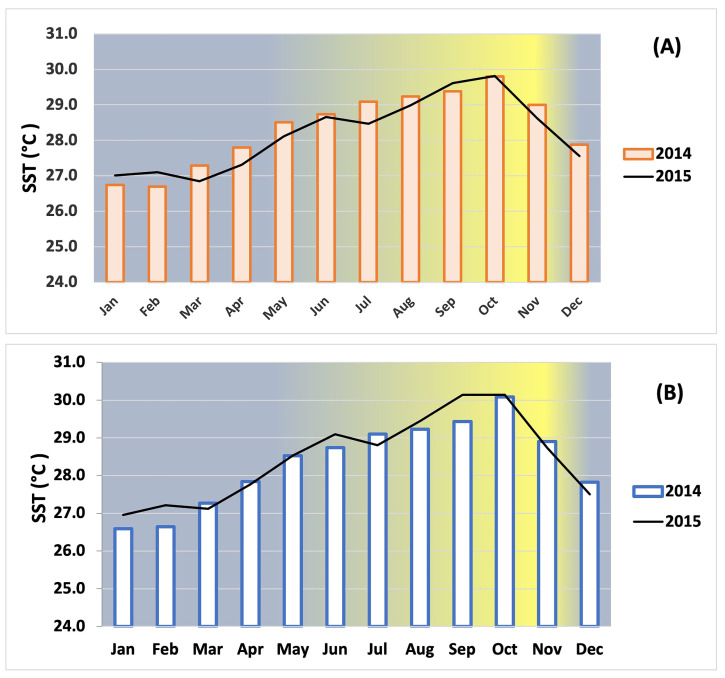
Monthly average sea surface temperatures (SST) during cold season (Jan–Feb) and warm season (Sep–Oct) at the inner-shelf reef Enrique (**A**) and at the mid-shelf reef San Cristobal (**B**), respectively, in La Parguera, Puerto Rico, during 2014 (square bars) and 2015 (black lines). The yellow in the gradient background indicates the highest temperature.

**Figure 2 proteomes-12-00020-f002:**
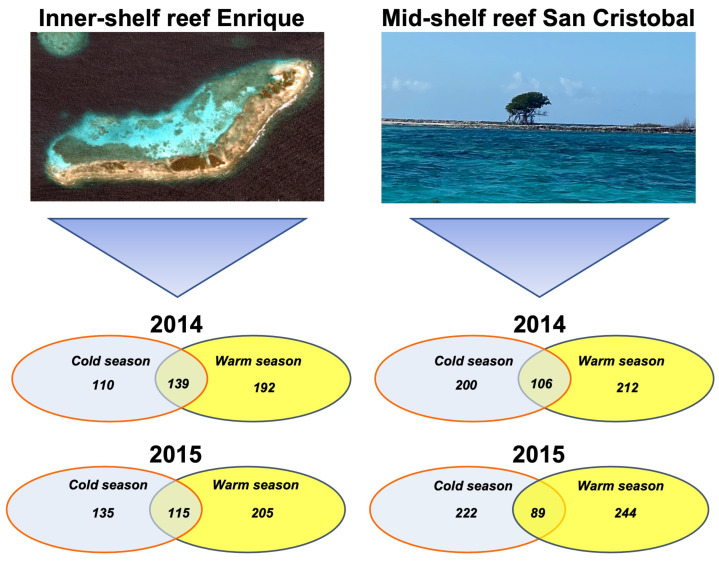
IKONOS satellite image from the inner-shelf reef Enrique and photo from the mid-shelf reef San Cristobal, with the corresponding Venn diagram indicating the differentially expressed proteoforms in 2014 and 2015 for *Orbicella faveolata* during the cold (blue circle) and warm seasons (yellow circle) in Puerto Rico.

**Figure 3 proteomes-12-00020-f003:**
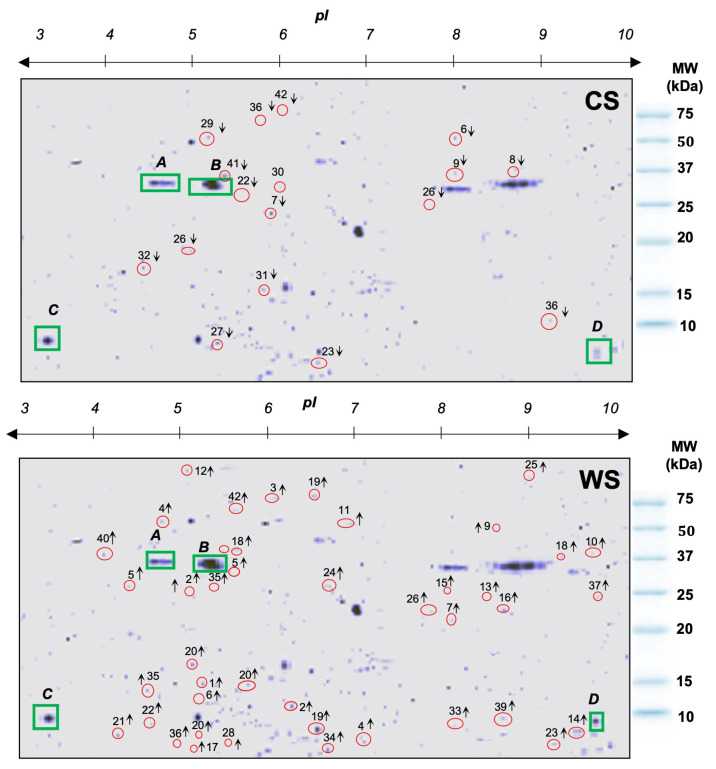
Two-dimensional gel electrophoresis showing differentially expressed proteoforms in *Orbicella faveolata* during cold season (CS) and warm season (WS). The top arrow from 3–10 (pI) represents the isoelectric point, defined as the pH at which the protein has no net charge, and the red circles with numbers from 1 to 42 on the 2D gels represent the proteoforms analyzed by MALDI-TOF/TOF. Common proteins are represented by green squares with capital letters A, B, C and D. ↑: Upregulated proteoforms, ↓: Downregulated proteoforms. Molecular Weight (MW) kilodalton (kDa).

**Figure 4 proteomes-12-00020-f004:**
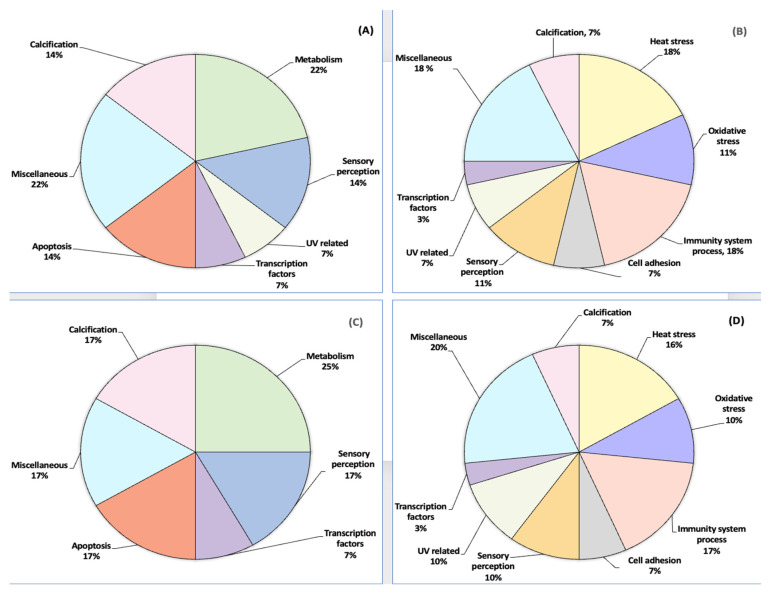
Pie representation of the proteoforms expressed according to their function in *Orbicella faveolata*. (**A**). Cold season at the inner-shelf reef Enrique, (**B**). Warm season at the inner-shelf reef Enrique, (**C**). Cold season at the mid-shelf reef San Cristobal, (**D**). Warm season at the mid-shelf reef San Cristobal.

**Figure 5 proteomes-12-00020-f005:**
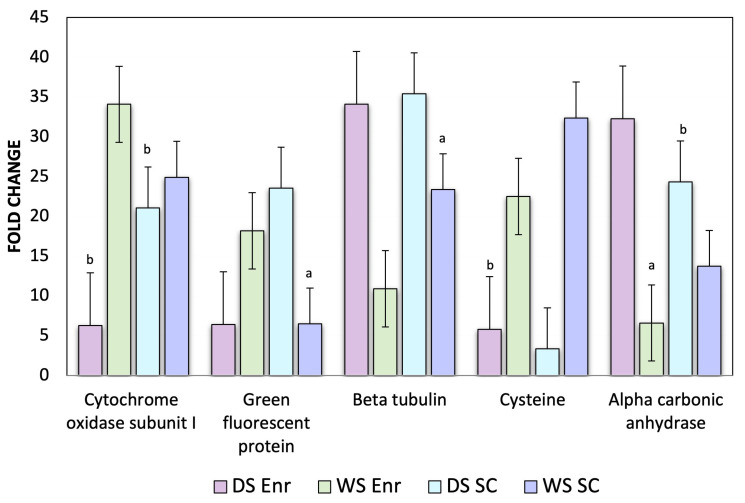
RT-PCR analysis conducted to validate the expression of five candidate genes in *Orbicella faveolata*. African violet bar: Cold season at the inner-shelf reef Enrique. African sage bar: Warm season at the inner-shelf reef Enrique. Powder blue bar: Cold season at the mid-shelf reef San Cristobal reef. Frosty lake bar: Warm season at the mid-shelf reef San Cristobal. “a” represents significantly up-regulated genes (*p* < 0.05) and “b” represents significantly down-regulated genes (*p* < 0.05).

**Figure 6 proteomes-12-00020-f006:**
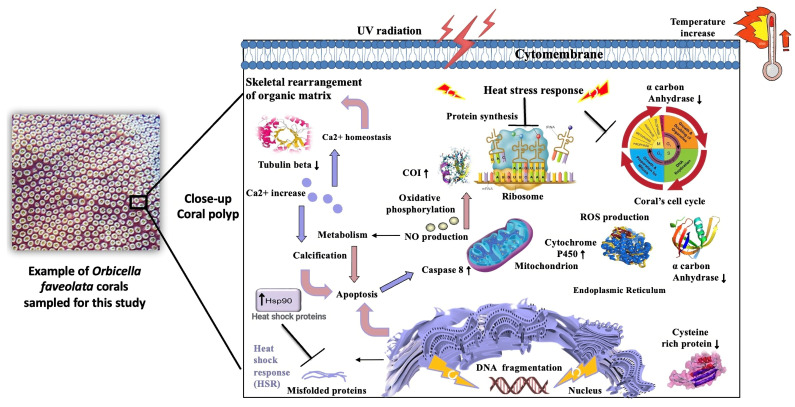
Diagrammatic representation of heat response to stresses, such as ROS (oxidative stress), DNA damage, or accumulation of misfolded proteins, heat shock response (HSR) affected by warm season in *Orbicella faveolata*. The proteoforms highlighted with up and down arrows, were identified through mass spectrometry and validated by RT-PCR. Proteins with up arrow and down arrow represent up-regulation and down-regulation.

**Table 1 proteomes-12-00020-t001:** qRT-PCR primers used to amplify five differentially expressed genes and one housekeeping gene in *Orbicella faveolata*.

RT-PCR Primers for *Orbicella faveolate*			
Gene	Forward Primer (5′–3′)	Reverse Primer (5′–3′)	Citation
Actin (Housekeeping gene)	GAGATGAAGGCTGCATCTGGAGAG	ACAAAGCTTGGCTGGAACA	[21]
Alpha carbonic anhydrase	ACAATTTTCGATGCAGGGCT	GCTCCGAACCTTGCGAATTA	[22]
Cytochrome c oxidase subunit I	GCGCGATGTTAGGTGATGAT	TTCCTGCGCCTTGTTCAAC	[23]
Green fluorescent protein	TCTGCCCCATGGTAAGTGTT	GGCTTGCCTTTTCCTTCTCC	
Beta-tubulin	TCAGAGAAGAATATCCCGACAGA	GGTGTGGTGAGCTTCAGAGT	
Scleractinian cysteine-rich protein	GGCTGCCAAGTTTCATTTGTG	CAACGTTCATTTGGTGGCAC	[24]

**Table 2 proteomes-12-00020-t002:** Differentially expressed proteoforms in *Orbicella faveolata*. Key: Acc. No.^a^: accession number, pI/Mw^b^: isoelectric point and molecular mass, P^c^: # of peptides, ^d^: MS score, EN^e^: Inner-shelf reef Enrique, f.c^f^: fold change, SC^g^: mid-shelf reef San Cristobal, f.c^h^: fold change, (+/−)^i^: Upregulated/downregulated in the Inner-shelf reef Enrique, and (+/−)^j^: Upregulated/downregulated in the mid-shelf reef San Cristobal. A 2-fold increase+/decrease− (+/−) cold season versus warm season in the inner-shelf reef Enrique and in the mid-shelf reef San Cristobal (EN, SC). Spot intensities was set as the threshold for indicating significant fold changes during the years 2014 and 2015.

Spot #	Category	Protein Name	Acc. No.^a^	GO Molecular Function	Theoretical pI/Mw^b^	P^c^	MS/MS^d^	EN^e^	SC^g^	EN (+/−)^i^	SC (+/−)^j^
								f.c^f^	f.c^h^		
1	Heat stress	Galaxin-like 1 [*Acropora millepora*]	ADI50284	Defense against oxidative stress	5.06/17.93	7	56	4.40	2.62	+	+
2		Heat shock transcription factor	ACH53605.1	DNA-binding	6.38/8.96	11	23	4.83	3.33	+	+
3		Heat shock protein 70 A1-like [*Orbicella faveolata*]	XP_020619238.1	Stress response	5.54/70.15	76	396	6.72	5.40	+	+
4		Heat shock protein HSP 90-beta-like [*Orbicella faveolata*]	XP_020618369.1	Protein folding	4.82/85.05	32	270	4.31	5.22	+	+
5		Endoplasmin-like [*Orbicella faveolata*]	XP_020606735.1	Protein folding and sorting	4.87/28.26	6	280	5.25	5.04	+	+
6	Metabolism	Cytochrome b [*Montastraea faveolata*]	Q762T3	Ubiquinol-cytochrome-c reductase	8.19/43.22	7	154	5.23	3.25	−	−
7		Cytochrome oxidase subunit I, partial (mitochondrion) [*Orbicella faveolata*]	AHA90940.1	Oxidative phosphorylation	5.54/21.89	11	280	2.06	2.06	−	−
8		Cytochrome P450 74A [*Acropora palmata*]	ACD42778.1	Oxidoreductase	8.99/40.80	3	123	1.86	1.93	−	−
9	Oxidative stress	Activin 1 protein [*Acropora digitifera*]	BAQ19091.1	Free radical binding	8.71/48.39	9	78	4.84	2.28	+	+
10		NADH dehydrogenase subunit 1	Q4G6D2	Oxidation-reduction	9.10/35.04	8	86	4.50	4.23	+	+
11		Thioredoxin reductase [*Acropora millepora*]	AFI99106.2	Oxidoreductase	6.37/74.30	5	88	2.33	2.45	+	+
12	Immune system process	Apextrin [*Acropora millepora]*	ABK63971.2	Innate immune response	5.08/96.1	10	22	3.20	3.20	+	+
13		ATP synthase F0 subunit 6 (mitochondrion) [*Orbicella faveolata*]	YP_271955.1	Hydrogen ion transmembrane transporter	8.46/25.07	7	38	2.60	2.56	+	+
14		ATP synthase subunit 8	Q8SJB2	Hydrogen ion transmembrane transporter	9.51/8.52	6	214	3.93	2.87	+	+
15		ATPase subunit 6 (ATP synthase subunit 6)	Q9TBW1	ATPase synthase subunit 6	7.92/25.37	11	87	15.45	15.43	+	+
16		Toll-like receptor 2 isoform X4 [*Orbicella faveolata*]	XP_020617122.1	Innate immune response	8.61/24.40	7	66	14.30	11.25	+	+
17	Cell adhesion	Mini-collagen, partial [*Acropora palmata*]	AAM74869.1	Collagen Trimer	5.6/6.18	8	465	9.22	8.89	+	+
18		2-acylglycerol O-acyltransferase 1-like [*Orbicella faveolata*]	XP_020602679.1	Acyltransferase	9.24/38.12	6	230	8.41	7.58	+	+
19	Sensory perception	Integrin subunit betaCn1 [*Acropora millepora*]	AAB66910.1	Cell matrix adhesion	6.41/87.84	7	254	1.60	2.47	+	+
20		Actin, partial [*Acropora millepora*]	ABY40470.1	Calcium regulation	5.26/17.33	11	489	6.83	5.87	+	+
21		Beta-tubulin, partial [*Orbicella faveolata*]	AHZ61618.1	GTPase	4.57/8.99	6	336	4.24	3.24	+	+
22		PREDICTED: unconventional myosin-VIIa-like [*Acropora digitifera*]	XP_012559373.1	Phagocytosis, engulfment	5.83/40.51	13	455	7.31	6.48	−	−
23		Myosin-IIIb-like [*Orbicella faveolata*]	XP_020624818.1	Calcium-regulation	6.82/3.27	4	325	4.62	3.45	−	−
24	UV related	GFP-like fluorescent chromoprotein cFP484 [*Orbicella faveolata*]	XP_020605513.1	Generation of precursor metabolites and energy	6.51/25.68	4	325	6.51	5.24	+	+
25		GPI inositol-deacylase-like [*Orbicella faveolata*]	XP_020615720.1	GPI anchored proteins	9.08/100.56	4	325	3.96	3.45	+	+
26		Green fluorescent protein [*Orbicella faveolata*]	ABC68475.1	Photoprotection	7.75/25.54	7	76	7.23	6.44	−	+
27	Transcription factors	PaxC, partial [*Acropora palmata*]	ACI29769.1	DNA binding	5.01/2.99	10	599	2.01	1.47	−	−
28		BMP2/4, partial [*Orbicella faveolata*]	AHZ61804	Growth factor	5.31/5.00	7	456	2.60	1.98	+	+
29	Apoptosis	Caspase 8 [*Acropora palmata*]	ADG23096.1	Programmed cell death	5.94/50.79	4	654	3.30	2.56	−	−
30		PREDICTED: ectonucleoside triphosphate diphosphohydrolase 2-like [*Acropora digitifera*]	XP_015779265	DNA binding	6.01/31.70	3	345	3.60	2.54	−	−
31	Calcification	Carbonic anhydrase 5A, mitochondrial-like [*Orbicella faveolata*]	XP_020613891.1	Carbonate dehydratase	5.81/15.74	4	180	6.40	5.34	−	−
32		Calmodulin-like protein, partial [*Acropora millepora*]	ACY07618.1	Calcium binding	4.74/19.66	3	259	8.40	9.26	−	−
33		Small cysteine-rich protein 8	B2ZG38	Neurotoxin	8.67/8.10	7	204	6.45	3.25	+	+
34		DnaJ homolog subfamily C member 5-like [*Orbicella faveolata*]	XP_020605856.1	Protein folding	6.9/8.26	4	45	9.23	3.42	+	+
35	Miscellaneous	Alpha i G protein [*Acropora palmata*]	AFZ78088.1	Signal transducer	5.72/40.35	9	573	9.22	9.20	+	+
36		Small cysteine-rich protein 6	C1KIZ5	Calcification	4.67/9.33	8	23	5.39	4.49	−	−
37		Zinc finger protein 768-like [*Orbicella faveolata*]	XP_020615964.1	Transcriptional regulation	9.59/25.29	6	26	2.64	2.51	+	+
38		Alpha q G protein [*Acropora palmata*]	AFZ78090	Polypeptide binding	5.32/41.88	6	736	2.31	2.31	+	+
39		Alpha carbonic anhydrase, partial [*Orbicella faveolata*]	AHZ61699.1	Carbonate dehydratase	8.24/11.20	4	456	7.46	6.45	+	+
40		Uncharacterized protein LOC110065766 [*Orbicella faveolata*]	XP_020628594.1	Uncharacterized protein LOC110065766 [*Orbicella faveolata*]	4.28/46.09	3	96	6.15	5.99	+	+
41		Galaxin-2-like [*Orbicella faveolata*]	D9IQ16.1	Calcium ion transport	5.45/35.12	8	126	8.03	7.25	−	+
42		Procollagen galactosyltransferase 1-like [*Orbicella faveolata*]	XP_020604034	Iron-ion binding	5.91/68.37	6	320	7.02	6.48	−	−

## Data Availability

Any data from this manuscript can be requested and is available upon request to CCD.

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
