# Peer review of "Seasonal Proteome Variations in Orbicella faveolata Reveal Molecular Thermal Stress Adaptations"

_proteomes, 2024, doi:10.3390/proteomes12030020_

Round 1

Reviewer 1 Report (Previous Reviewer 1)

Comments and Suggestions for Authors

I appreciate that you considered my suggestions. The manuscript has been modified and I agree with the changes made.

Author Response

Dear Reviewer 1:

Your recognition of our work and encouraging comments are greatly appreciated.

Thank you!

Reviewer 2 Report (New Reviewer)

Comments and Suggestions for Authors

Ricarute et al., present a proteome analysis of Orbicella faveolata that highlights the changes in protein expression due to seasonal temperature variations. They demonstrated that even with a temperature change of only 4°C between cold and warm seasons, multiple proteins are differentially regulated in O. faveolata. Specifically, proteins involved in heat stress and oxidative stress are upregulated during the warm season suggesting that these proteins help O. faveolata to adapt to higher temperatures. The data and the analysis are convincing. Furthermore, understanding how coral reefs adapt to warmer temperatures due to global warming is crucial. I indeed support the publication in Proteome but the authors need to address some points first.

Main points

Introduction

Could the authors summarize previous studies on Orbicella faveolata and position their study within this context? Specifically, what is new in their research and what has already been done regarding the changes in temperature?

Results

-The choice of validating mass spectrometry using qPCR needs to be addressed in the paper. RNA levels can differ significantly from protein levels. Why did the authors choose qPCR instead of western blotting? The authors need to comment on this (are antibodies not available?).

-In the limitations of the study, the authors mention that numerous proteins found in O. faveolata have unknown functions. What is the percentage of unknown proteins? What is the percentage of unknown proteins for each category in the Venn diagram in Figure 2?

-Can the authors describe in the Materials and Methods in details how the colonies were tagged?

In the limitations of the study, the authors mention that numerous proteins found in O. faveolata have unknown functions. What is the percentage of unknown proteins? What is the percentage of unknown proteins for each category in the Venn diagram in Figure 2? Can the authors describe in the Materials and Methods how the colonies were tagged?

Minor points

Text

Please go through the text and ensure consistency (e.g., p-value should be consistently formatted). The Celsius symbol is not formatted correctly throughout the paper. Additionally, there are many typos. Here are some examples:

  • Line 127: Fix the typo "]]".
  • Figure 1: The bottom y-axis needs a closing parenthesis.
  • Line 111: Change "distaining" to "destaining".
  • Line 137: "A mass" is repeated twice.
  • Line 207: "GO: gene ontology" (the word "gene" is missing).

These are just a few examples. Please review the entire document for similar issues.

Figures

I believe that improving the figures is necessary to facilitate the understanding of the paper.

  • Define "ip" in the legend of Figure 3.
  • Figure 4: Please change the combination of colors. They are very hard to see and very bright. Ensure consistency in color use for the same classes. For example, heat stress in panel B is yellow with a pattern, but heat stress in panel D has no pattern.
  • Figure 5: Please change the colors and make the background white.
  • Figure 6: Please simplify. There are too many arrows, making the figure too hard to understand.

Author Response

Comments and Suggestions for Reviewer 2:

Ricaurte et al., present a proteome analysis of Orbicella faveolata that highlights the changes in protein expression due to seasonal temperature variations. They demonstrated that even with a temperature change of only 4°C between cold and warm seasons, multiple proteins are differentially regulated in O. faveolata. Specifically, proteins involved in heat stress and oxidative stress are upregulated during the warm season suggesting that these proteins help O. faveolata to adapt to higher temperatures. The data and the analysis are convincing. Furthermore, understanding how coral reefs adapt to warmer temperatures due to global warming is crucial. I indeed support the publication in Proteome but the authors need to address some points first.

Main points

Introduction

Could the authors summarize previous studies on Orbicella faveolata and position their study within this context? Specifically, what is new in their research and what has already been done regarding the changes in temperature.

Author’s Response:

A paragraph addressing this within the manuscript, was added in page 2, lines 56-58 as follows: Previous studies indicated that ambient and elevated water temperatures cause hypoxia in O. faveolata, reducing oxygen consumption [12]. Additionally, researchers have used artificial intelligence to predict bleaching in O. faveolata based on protein signatures at higher temperatures [13]. Further studies examining the proteomes of both coral hosts (Orbicella faveolata) and their endosymbiotic dinoflagellates under different laboratory temperatures found increased protein turnover at higher temperatures [14]. However, understanding O. faveolata coral differential protein expression response and how it adapts to warmer seasonal temperatures conditions (from cold to warm seasons) in two different reefs: the inner-shelf reef Enrique and the mid-shelf reef San Cristobal, southwest Puerto Rico, in the Caribbean has not been studied.

Reviewer 2:

Results

The choice of validating mass spectrometry using qPCR needs to be addressed in the paper. RNA levels can differ significantly from protein levels. Why did the authors choose qPCR instead of western blotting? The authors need to comment on this (are antibodies not available?).

Author’s Response: RT-qCR is one of the most widely used technologies to validate proteomics data at the gene expression levels as our publication and other studies indicate (Ricaurte et al, 2016, Gaurav et al, 2016: Evaluation and validation of reference genes for qPCR analysis to study climate change-induced stresses in Sinularia cf. cruciata (Octocorallia: Alcyonidae), Mayfield et al, 2020, Proteomic Signatures of Corals from Thermodynamic Reefs. Microorganisms. 2020 Aug 1;8(8):1171. doi: 10.3390/microorganisms8081171. PMID: 32752238; PMCID: PMC7465421…). Our mass spectrometry-based proteomics data detected and quantified proteins, but it did not directly measure mRNA levels. Therefore, qPCR was chosen to verify if changes in protein expression levels were attributable to changes in gene expression, thereby enhancing our understanding of the biological processes involved as depicted in the proposed Stress response pathway (Figure 6). Another factor influencing our choice of qPCR was its sensitivity, rapid results, and the user-friendly equipment available in our marine science department, unlike western blot, which can only measure relative protein abundance. For O. faveolata, our intention is to use qPCR to ensure that observed changes in mRNA levels correspond to changes in protein expression in response to seasonal fluctuations in temperature.

Reviewer 2:

Can the authors describe in the Materials and Methods in details how the colonies were tagged?

Author’s Response:

 The following paragraph was added in the manuscript, Page 2, lanes 78-81 as follows:

Coral colonies were tagged with small plastic tags imprinted with a unique identifying number -coded "cattle tagattached to a 3-inch hardened masonry nail that has been driven into the substrate near the colony. This allowed for the identification and location of the specific colonies during 2014 and 2015 (See Supplementary Table S1 and S4).

Reviewer 2: In the limitations of the study, the authors mention that numerous proteins found in O. faveolata have unknown functions. What is the percentage of unknown proteins? What is the percentage of unknown proteins for each category in the Venn diagram in Figure 2?

Author’s Response: In the pie chart, unknown proteins were omitted for several reasons: 1. These proteins did not meet the selection criteria as they exhibited less than a 2-fold change. 2. Including unknown functions could potentially obscure the chart and lead to misinterpretation of the data. 3. Our focus was to emphasize significant proteins whose known functions could provide relevant insights into changes in temperature-related upregulation or downregulation and, 4. By excluding unknown proteins, we aimed to enhance the comparability among the known proteins.

Reviewer 2: Minor points

Text

Please go through the text and ensure consistency (e.g., p-value should be consistently formatted). The Celsius symbol is not formatted correctly throughout the paper. Additionally, there are many typos. Here are some examples:

Line 127: Fix the typo "]]".

Figure 1: The bottom y-axis needs a closing parenthesis.

Line 111: Change "distaining" to "destaining".

Line 137: "A mass" is repeated twice.

Line 207: "GO: gene ontology" (the word "gene" is missing).

These are just a few examples. Please review the entire document for similar issues.

Author’s Response:

All document was reviewed, and typos were corrected, ensuring consistency in formatting.

Reviewer 2:

Figures

I believe that improving the figures is necessary to facilitate the understanding of the paper.

Define "ip" in the legend of Figure 3.

Figure 4: Please change the combination of colors. They are very hard to see and very bright. Ensure consistency in color use for the same classes. For example, heat stress in panel B is yellow with a pattern, but heat stress in panel D has no pattern.

Figure 5: Please change the colors and make the background white.

Figure 6: Please simplify. There are too many arrows, making the figure too hard to understand.

Author’s Response:

All Figures 1 through 6 were modified according to the suggestions.

  • "ip" was defined in the legend of Figure 3 as follows: The isoelectric point (pI) is defined as the pH at which the protein has no net charge.
  • Figure 4: We thank reviewer 2 for this suggestion, we subsequently changed the combination of colors to less bright colors and the heat stress category in panel B is now yellow with no pattern to uniformize it and be consistent in color use with the panel D yellow color.
  • Figure 5: We changed the colors to less bright ones and made the background white.
  • Figure 6 was optimized and mechanistically targeted on Orbicella faveolata’s proteomics heat responseto stresses, such as ROS (oxidative stress), DNA damage, or accumulation of misfolded proteinsheat shock response (HSR) to ensure clarity and enhance understanding.

This manuscript is a resubmission of an earlier submission. The following is a list of the peer review reports and author responses from that submission.

Round 1

Reviewer 1 Report

Comments and Suggestions for Authors

This is an interesting approximation study to understand the response of the coral to thermal stress in these particular locations, although the temperature remains within the range of a tropical zone, without such a marked variation between seasons, the results reflect a variation in the expressed proteins which tells us about the adaptation processes of this species.

Regarding my observations on the manuscript, I consider that organization is needed in the presentation of results since some results figures in the methodology and discussion section should be move to results.

When editing, only consider the spacing between words, since there are double spaces in various document parts.

Author Response

Thank you very much, Reviewer #1 for agreeing with us to the intention of this manuscript and finding the results of our manuscript interesting by adding to our knowledge of phenotypic plasticity and acclimatization mechanisms in corals. Secondly, following your advice which indeed improved the manuscript's presentation and display of the results, we have: 

 moved figure 1 and figure 3 that were initially in the methodology and in the discussion to the results’ section of the current version of the manuscript. We also carefully reviewed the results organization and checked the double-spacing accordingly.

Reviewer 2 Report

Comments and Suggestions for Authors

Major problems:

Line 110-112: “To control for gel-to-gel variation in protein loading across gels, the spot volume density obtained from each individual lung lysate was calculated               by image analysis software and normalized to total spot volume on that gel.” Corrals have no LUNGS, its copy-paste from Kelsen, Duan, Ji, et al.: Cigarette Smoke Induces an Unfolded Protein Response. Am J Respir Cell Mol Biol Vol 38. pp 541–550, 2008

Several method descriptions are incomplete.

Samples were collected 2014-2015; authors provide no information on how the temperature data were collected, the “monthly average sea surface temperature” is presented on figure 1, the difference between “cold” and  ”warm” season in 3.1°C. However, in the abstract, discussion and conclusions authors speculate about “temperature seasonality fluctuations”, “heat tolerance, acclimatization mechanism during temperature fluctuations”, “accelerated climate change mechanisms underlying thermal adaptation in corals”.

2DE results are inaccurately presented: Criteria for protein identification are missing, the date of Swissport database access is missing, some protein accession numbers are not present in any public database, organism names for half of the proteins are missing in the table 2.  Not clear what authors mean by up-and down-regulation (i.e. in comparison to what?). Moreover, fold changes for protein abundance are presented in Table 2, however no p-values are presented, thus its impossible to asses reproducibility/reliability of the results. Not clear how from ”580 differentially expressed proteoforms in both seasons” authors selected 42 proteins.

2DE results are over-interpretated and the title/conclusions do not correspond to experiments and results. The fact that 42 proteins were classified as belonging to 11 biological processes (according to public databases) does not automatically means “reveal acclimation mechanisms”.   

No any reference to previously published (and very similar) Mayfield et al. 2021 https://doi.org/10.3389/fmars.2021.660153

Author Response

Reviewer 2: Line 110-112: “To control for gel-to-gel variation in protein loading across gels, the spot volume density obtained from each individual lung lysate was calculated by image analysis software and normalized to total spot volume on that gel.” Corrals have no LUNGS, its copy-paste from Kelsen, Duan, Ji, et al.: Cigarette Smoke Induces an Unfolded Protein Response. Am J Respir Cell Mol Biol Vol 38. pp 541–550, 2008

Author’s Response: Proteomics techniques are used in a variety of systems and like most molecular procedures they have been developed first in questions regarding humans or a handful of model species.  Such a citation should not be unusual. We will not comment on the reviewer’s remark that corals have no lungs. However, we want to thank him for pointing out our omission to paraphrase the sentence about how to control gel to gel variation. We have now edited the sentence to:

In order to improve the correctness and quantification of protein levels from different gels, we normalized the spot volume density attained from each coral lysate against the total spot volume of that particular gel with the aid of PDQuest software as cited from our previous publication (Ricaurte et al, 2016).

Reviewer 2: Several method descriptions are incomplete.

Author’s Response:

Method descriptions have been carefully verified, completed, and updated as necessary.

Reviewer 2: Samples were collected 2014-2015; authors provide no information on how the temperature data were collected, the “monthly average sea surface temperature” is presented on figure 1, the difference between “cold” and ”warm” season in 3.1°C. However, in the abstract, discussion and conclusions authors speculate about “temperature seasonality fluctuations”, “heat tolerance, acclimatization mechanism during temperature fluctuations”, “accelerated climate change mechanisms underlying thermal adaptation in corals”.

Author’s Response: We thank reviewer 2 for pointing this out. We now specified how SST was measured in Lane 78, page 2, under Materials and Methods as follows: The monthly average sea surface temperatures (SST) were measured every hour in situ with a Hobo-pro during cold season (Jan-Feb) and warm season (Sep-Oct) of 2014-2015.

We also clarified in the abstract that indeed the temperature difference of 3.1 oC between warm and cold seasons in both reefs (the inner-shelf reef Enrique and the mid-shelf reef San Cristobal), affects the expression of proteins associated with stress response.

Corals are extremely sensitive to temperature changes. The warming of 1-2 degree Celsius in water temperature is enough to trigger heat stress protein expression.

The UN Intergovernmental Panel on Climate Change projects that at just 2 degrees C of warming, 99 percent of corals would be lost.Feb 15, 2023:

https://e360.yale.edu/digest/pacific-coral-reefs-adapting-to-warmer-waters-study-finds#:~:text=Some%20corals%20in%20the%20eastern,of%20corals%20would%20be%20lost.

Reviewer 2:

2DE results are inaccurately presented: Criteria for protein identification are missing, the date of Swissport database access is missing, some protein accession numbers are not present in any public database, organism names for half of the proteins are missing in the table 2. Not clear what authors mean by up-and down-regulation (i.e., in comparison to what?). Moreover, fold changes for protein abundance are presented in Table 2, however no p-values are presented, thus it is impossible to assess reproducibility/reliability of the results. Not clear how from ”580 differentially expressed proteoforms in both seasons” authors selected 42 proteins.

Reviewer 2: 2DE results are inaccurately presented: Criteria for protein identification are missing, the date of Swissport database access is missing, some protein accession numbers are not present in any public database, organism names for half of the proteins are missing in the table 2. Not clear what authors mean by up-and down-regulation (i.e., in comparison to what?). Moreover, fold changes for protein abundance are presented in Table 2, however no p-values are presented, thus its impossible to assess reproducibility/reliability of the results. Not clear how from ”580 differentially expressed proteoforms in both seasons” authors selected 42 proteins.

Author’s Response: The criteria for protein identification were assessed via the search engine Mascot, developed by Matrix Science and Protein Pilot search engine parameters against the 2015 Swissprot database with the species set as corals. This was added to the paragraph of the manuscript (page 2, line 72 as: Database searching was accomplished using the MatrixScience website (http://www.matrixscience.com/xmlns/schema/mascot_search_results) for MS and MS/MS interrogations on coral proteins and Protein Pilot search engine parameters against Swiss-Prot databank (www.expasy.org). Proteins 22 and 23  were removed from Genebank as a result of standard genome annotation processing in the gene bank. The accession numbers and protein names were adjusted as indeed needed and requested by reviewer 2. The fold change analysis is presented as Supplemental material 2. As reviewer 2 highlights that 580 differentially expressed proteoforms were detected in both seasons, and questions why only 42 proteins were selected.  The reasons behind are as follows: 1) Because it was not feasible to sequence all 580 resolved proteins, the selection criteria for sequencing were based on the quantitative analysis of the 2D gel image (Figure 3) with the goal of obtaining a broad scope of the proteins differentially expressed, we selected a total of 42 protein spots as the most significantly differentially expressed by fold change ≥ 2, P ≤ 0.05 , as calculated by PDQuest™ software. 2) A 2-fold increase+/decrease- (+/-) cold season versus warm season in both reefs (the inner-shelf reef Enrique and the mid-shelf reef San Cristobal ENe, SCg, EN, SC). Spot intensities was set as the threshold for indicating significant fold changes during the years 2014 and 2015. A 2-fold increase+/decrease- (+/-) cold season versus warm season in the in the inner-shelf reef Enrique and in the mid-shelf reef San Cristobal (EN, SC). Spot intensities was set as the threshold for indicating significant fold changes during the years 2014 and 2015.−, decrease, + increase, see table 1. p-values are in the Supplementary material 1, under the statistical analysis.

To address the reviewer question on: Not clear what authors mean by up-and down-regulation (i.e., in comparison to what?).

Author’s Response: The upregulated differentially expressed proteins in the inner-shelf reef Enrique were were overexpressed when Compared With Cold Season (CWCS),while the downregulated had a lower expression when Compared With Warm Season (CWWS) in that same reef.  In the mid-shelf reef San Cristobal, the upregulated differentially expressed proteins were overexpressed when Compared With Cold Season (CWCS),while the downregulated had a lower expression when Compared With Warm Season (CWWS) in that same reef. Abbreviations were clarified accordingly:Compared With Cold Season (CWCS) and Compared With Warm Season (CWWS).

Reviewer 2: 2DE results are over-interpretated and the title/conclusions do not correspond to experiments and results.

Author’s Response: We took in consideration reviewer’s 2 comment and subsequently revised accordingly and changed the title to (Seasonal proteome variations in Orbicella faveolata reveal molecular thermal adaptations) for a better description of our research. Moreover, the conclusions were further clarified as aligned with the proposed molecular mechanism (Figure 6) (New manuscript: Page 17, lines 434). 

 Reviewer 2: (according to public databases) does not automatically means “reveal acclimation mechanisms”. No any reference to previously published (and very similar) Mayfield et al. 2021. 

Author’s Response: We are grateful to Reviewer 2  for this insightful suggestion that helped us improve this manuscript.  We subsequently refrained from linking possible role of acclimation based on public databases and cited as suggested Mayfield et al. 2021 (as indeed a valuable reference to our manuscript) by adding the following paragraph in the Discussion, starting as the last paragraph of Page  13 and 6 lines into Page 14.

Mayfield and colleagues reported similar results on stress-protective proteins including those associated with thermotolerance, protein folding, calcification functions and response to oxidative stress in O faveolata, in a thermal laboratory-based setting mimicking conditions from inshore and offshore localities in south Florida. Interestingly, our results across experiments showed that temperature conditions (from cold and warm seasons) in both reefs: the inner-shelf reef Enrique and the mid-shelf reef San Cristobal, southwest Puerto Rico, also led to key thermotolerance signature proteins that can be a valuable tool for future comparative studies on the identification of proteins and mechanisms, that O faveolata employs to mitigate thermal stress. 

We also updated Table 2 as requested by reviewer 2, and agree that some proteins did not have the complete names and accession numbers in the gene bank. This valuable observation from reviewer 2 helped us to improve Table 2 with the corresponding accession number for the proteins. We subsequently included the Standard Protein BLAST peptide table that includes the higher score in comparison with Orbicella specie (Supplemental Table S3).

We would like to thank again reviewer 2 for the thoughtful comments and efforts towards improving our manuscript.

Reviewer 3 Report

Comments and Suggestions for Authors

The authors performed a comparative proteomics study between cold and warm seasons to investigate how the coral Orbicella faveolate changes their proteoform profiles to copy with seasonal fluctuations of temperature. It is an interesting topic, and numerous data of 2D-gel electrophoresis, MS and qRT-PCR were generated to support the main conclusions. The overall writing of this manuscript is generally good. However, as the authors mentioned in lines 74-76, the coral colonies were collected at depths of 1-3 m, while they measured monthly average sea surface temperatures (SST). It is very strange that these parameters, in fact, couldn’t reflect the exact environmental temperatures for the corals’ growth. Therefore, the proposed connection between SST seasonality fluctuations and acclimation mechanisms (lines 2-3) is unreasonable and unbelievable. The present version of this manuscript is unacceptable; the authors are recommended to rewrite it with reasonable descriptions and discussions (although the data are reasonable).

Author Response

Reviewer 3:

The authors performed a comparative proteomics study between cold and warm seasons to investigate how the coral Orbicella faveolate changes their proteoform profiles to copy with seasonal fluctuations of temperature. It is an interesting topic, and numerous data of 2D-gel electrophoresis, MS and qRT-PCR were generated to support the main conclusions. The overall writing of this manuscript is generally good.

 However, as the authors mentioned in lines 74-76, the coral colonies were collected at depths of 1-3 m, they measured monthly average sea surface temperatures (SST). It is very strange that these parameters, in fact, couldn’t reflect the exact environmental temperatures for the corals’ growth. Therefore, the proposed connection between SST seasonality fluctuations and acclimation mechanisms (lines 2-3) is unreasonable and unbelievable. The present version of this manuscript is unacceptable; the authors are recommended to rewrite it with reasonable descriptions and discussions (although the data are reasonable).

Author’s Response:

We thank reviewer 3 on the interest in our manuscript. While we appreciate the reviewer's feedback questioning the proposed connection between SST seasonality fluctuations and acclimation mechanisms, we respectfully would like to clarify that seasonal water temperatures typically fluctuate less than 4°C across the majority of tropical reefs (Donner, 2011; Lough, 2012), 1°Celsius being considered sufficient to alter coral growth and metabolism. Analyses of coral bleaching on Caribbean reefs over the past two decades suggests that small increases in regional sea surface temperature (0.1 °C) is enough to link this small difference to adaptation/acclimatation’s mechanisms in coral. This has been abundantly demonstrated by other authors (Brown 1997; Hoegh-Guldberg 1999; Hughes et al. 2003; Baker et al. 2008; Donner et al. 2007, Donner, 2011; Lough, 2012). We nevertheless refrained from drawing direct conclusions on the acclimation in the current version of the manuscript. We respectfully disagree that colonies from shallow water (1–3 m depth) do not reflect coral environmental temperatures for the corals’ growth since numerous publications including ours refers to O. faveolata as being collected from the reef between 2 and 8 m depth by SCUBA diving, where the Hobo temperature loggers were deployed in different reef habitats at two depth intervals (0-3 and 3-8m) and Levitan 2014, collected orbicella coral changes in SST at 3 to 8 m in depth as well as Adjeroud, et al 1999, Alvarado-Chacón, et al 2020,  Ruiz-Diaz et al 2020) and Weil and colleagues (Weil, 2009: Spatial and temporal variability in juvenile coral densities, survivorship and recruitment in La Parguera, southwestern Puerto Rico) within the two inner shelf coral reefs off La Parguera Natural Reserve (San Cristóbal and Enrique) in southwest, Puerto Rico, the same locations and depth as mentioned by our current study by co-author Weil and other authors Gomez et al, 2020: Comparison of Satellite-Based Sea Surface Temperature to In Situ Observations Surrounding Coral Reefs in La Parguera, Puerto Rico. Moreover, colonies from both shallow and deep water depth have been collected from their natural habitat without significant physiological differences in the coral population environments as evidenced (Martinez, et al, 2021). While we recognize that other environmental factors beside the temperature affect coral growth, nevertheless this was beyond the scope of this study that primarily focused on the seasonal temperature fluctuations in orbicella faveolata coral in that specific region at la Parguera, southwest. PR.  This work fit a model that can effectively estimate reef-depth (1-3m) water temperature from SST estimated by satellite remote sensing as observed in previously published studies (Gomez et al., 2020: Comparison of Satellite-Based Sea Surface Temperature to In Situ Observations Surrounding Coral Reefs in La Parguera, Puerto Rico) https://www.mdpi.com/2077-1312/8/6/453.  Additionally, in this current study, sea surface temperatures were monitored every hour with the hobo in-situ in the area where the colonies were extracted. Degree Heating Weeks (DHWs), defined as numbers of weeks during which SSTs exceed 1°C above the local month, have been shown to cause significant coral bleaching.  Moreover,  at only 1–2°C, the vital symbiosis of coral with dinoflagellate algae  breaks down in a process referred to as coral bleaching (Berkelmans and Willis 1999).

We thank reviewer 3 for his constructive and helpful comments, which have significantly improved the quality and clarity of our manuscript.

Round 2

Reviewer 3 Report

Comments and Suggestions for Authors

The authors made careful revisions in accordance with both reviewers' comments. The present version of this manuscript is acceptable.